# Field Work in Papua New Guinea Documents Seven New Records of a Hemiepiphytic Habit in Ferns

**DOI:** 10.3390/plants13081104

**Published:** 2024-04-15

**Authors:** Michael Sundue, Heveakore Maraia

**Affiliations:** 1Royal Botanic Garden Edinburgh, 20a Inverleith Row, Edinburgh EH3 5LR, UK; 2Biology Center of the Czech Academy of Sciences, Institute of Entomology, Branišovská 1160/31, 370 05 České Budějovice, Czech Republic; 3Department of Ecosystem Biology, Faculty of Science, University of South Bohemia, Branišovská 1645/31a, 370 05 České Budějovice, Czech Republic

**Keywords:** epiphyte, morphological convergence, plant habit, New Britain

## Abstract

Hemiepiphytes have captured the attention of biologists since they seemingly hold clues to the evolution of epiphytes themselves. Hemiepiphytes are known to occur sporadically in the leptosporangiate ferns, but our understanding of their evolution remains limited by the relatively small number of detailed observations. This study adds to our knowledge by documenting seven species previously assumed to be holoepiphytes. This finding was based on fieldwork conducted in the Baining Mountains of Papua New Guinea that resulted in 319 collections representing 206 species. Approximately 3% of these species were hemiepiphytes: *Asplenium acrobryum*, *A. amboinense*, *A. scandens*, *A. scolpendropsis*, *Crepidomanes aphlebioides*, *Leptochilus macrophyllus*, and *Sphaerostephanos scandens*. All started growth as low-trunk epiphytes, and later, as larger climbing plants, exhibited strongly dimorphic roots consisting of short clasping ones that affixed the rhizome to the trunks and long feeding roots that entered the soil. Most of the seven hemiepiphyte species that we found exhibited distichous phyllotaxy and dorsiventrally flattened rhizomes, suggesting morphological convergence associated with this habit in four families. These new records suggest that large hemiepiphytic clades occur in *Asplenium* and *Leptochilus*. Our observations expand the geographic and taxonomic breadth of hemiepiphytic ferns, provide a baseline estimate of their diversity within a tropical flora, and offer morphological and phylogenetic clues to uncover additional records.

## 1. Introduction

Most ferns species are terrestrial understory perennial herbs, but a variety of other habits are known, including lithophytic, arborescent, vines, scandent disturbance colonizers, and epiphytes [1,2]. Epiphytes are particularly well developed in ferns; ferns comprise 3% of the world’s vascular flora, but nearly 10% of all epiphytes [3]. Hemiepiphytes also occur in the ferns. They begin life as low-trunk epiphytes [4] but later connect to the ground via roots [5,6]. The recognition of hemiepiphytes in ferns dates back at least 20 years, probably longer, but many early authors did not apply the term as it is currently understood, and those records need to be reassessed [7,8,9,10,11,12,13]. Nitta and Epps [14] established a new wave of publications in their study of *Vandenboschia collariata* (Bosch) Ebihara & K. Iwats. by explicitly documenting its life history starting as gametophytes establishing on trunk bases and later connecting to the soil via roots. This inspired additional studies, and within the last fifteen years, about 10 new examples have been documented. In many of these cases, a holoepiphytic (on trees, disconnected from the ground) habit had been previously assumed, and the hemiepiphytic habit was only determined through careful field study [5,15,16,17]. In others, the habit was noted in the description of a new species [18,19,20,21]. These new records have captured the attention of biologists since they seemingly hold clues to the evolution of epiphytes themselves, with recent investigations asking whether hemiepiphytes have evolved from terrestrial, epiphytic, or epipetric ancestors [22,23,24,25]. To date, all three of these evolutionary histories have been supported. To further explore this topic, we searched for new hemiephytic records in Papua New Guinea, an understudied biodiversity hotspot [26]. 

## 2. Results

We found seven species of hemiepiphytes: *Asplenium acrobyrum*, *A. ambioinsene*, *A. scandens*, *A. scolopendropsis*, *Crepidomanes aphlebioides*, *Leptochilus macrophyllus*, and *Sphaerostephanos scandens*. In each species we observed the second and third criteria (see Section 4) used to diagnose them as hemiepiphytes; namely, that they make root contact with the soil, and the rhizomes are only found upon trees—never on the forest floor (Table 1). 

In both *Asplenium amboinsense* and *A. scolopendropsis*, we also confirmed that gametophytes established upon the phorophyte (criterion number one). We observed root dimorphism in each of these species, which is one of the morphological traits we expect to observe in hemiepiphytes. All our hemiepiphytic species except *A. scandens* had distichous phyllotaxy (Table 2). Dorsiventrally flattened rhizomes were observed in *A. amboinense*, *A. scandens*, and *Leptochilus macrophyllus*; the rhizomes were terete or nearly so in *Asplenium scolopendropsis, Crepidomanes aphlebioides*, and *Sphaerostephanos scandens*. While all ferns exhibit, to some degree, heteroblastic leaf sequences from small to large leaves, we observed an abrupt transition from small to large in *A. scolopendropsis* after roots had contacted the soil. *Crepidomanes aphlebioides* exhibited leaf dimorphy, but we are uncertain of the relation between leaf dimorphy and soil contact of the roots in that species. 

Our expedition to the Baining Mountains resulted in 319 collections representing 206 species of ferns and lycophytes. Of these species, seven are hemiepiphytes; thus, this life form comprises at least 3% of the fern and lycophyte species diversity in the Baining Mountains. In contrast, 43 species (~21%) were interpreted as holoepiphytes. The remaining 156 (76%) were terrestrial with a variety of habits, including climbers, scandent species, and tree ferns. 

### 2.1. Enumeration of Species

#### 2.1.1. *Asplenium acrobryum* Christ. (Aspleniaceae)

Observations—*Asplenium acrobryum* (Figure 1A) is a short-creeping low-trunk hemiepiphyte with large simple arching leaves (Figure 1B) that spread away from the phorophyte. The rhizome bears dorsal leaves (Figure 1D) and lateral roots (Figure 1D). The roots are dimorphic, and both types of roots, clasping and feeding (Figure 1E), are abundant. The long feeder roots branched profusely upon contacting the soil (Figure 1F), but not before then. The leaves also bear subapical proliferous buds (Figure 1C).

Vouchers—East New Britain Province: Wild Dog Camp, former site of SINIVIT Wild Dog Mine, Bootsiqui trail, −4.62616 152.04406, 1012 m, *Sundue & Maraia 4250* (BISH, LAE, VT). Madang Province: Madang, Bundi, −5.759149 145.235891, 1800 m *Sundue 3773* (LAE, VT, UC). Madang Province: Madang, Bundi, −5.759149 145.235891, 1800 m *Sundue 3775* (LAE, VT, UC).

#### 2.1.2. *Asplenium amboinense* Willd. (Aspleniaceae)

Observations—*Asplenium amboinense* (Figure 2A) juvenile plants (Figure 2B) were observed establishing as low-trunk epiphytes. Rhizomes are strongly dorsiventral with dorsal leaves and lateral roots (Figure 2D,E). The roots are conspicuously dimorphic, with short clasping roots (Figure 2E), and elongate feeder roots (Figure 2D). The simple and entire leaves spread from the phorophyte (Figure 2B,C) and bear proliferous buds at the lamina apices (Figure 2A).

Vouchers—Northern Province: Akupe Camp, Kuriae River, Sibium Mountains, Umate Village., −9.285825 148.27266, 740 m, *James & Sundue 1519* (BISH, LAE, VT). Madang, Bundi; −5.758982, 145.186093, 2200 m, *Sundue 3825* (LAE, VT, UC).

#### 2.1.3. *Asplenium scandens* J. Sm. (Aspleniaceae)

Observations—*Asplenium scandens* is a long-creeping low-trunk hemiepiphyte with divided leaves (Figure 3A,F) that spread away from the phorophyte (Figure 3B). The leaves are held in distichous arrangement, but close observation indicates that the phyllotaxis develops as helical. The rhizome is prominently dorsiventral (Figure 3D), with dorsal leaves and lateral roots (Figure 3D). The roots are sparsely produced, but strongly dimorphic with both clasping and feeding roots (Figure 3E).

Vouchers—East New Britain Province: Wild Dog Camp, former site of SINIVIT Wild Dog Mine, slopes above stream, −4.62616 152.04406, 1012 m, *Sundue & Maraia 4333* (BISH, LAE, VT). Madang. Bundi, 5°45′32.3958″S, 145°14′9.2076″E, 1800 m, *Sundue 3786* (LAE, VT, UC)

#### 2.1.4. *Asplenium scolopendropsis* F. Muell. (Aspleniaceae)

Observations—*Asplenium scolopendropsis* gametophytes and juvenile plants (Figure 4A–D) were observed establishing on the lower portions of trunks. Rhizomes were long-creeping, with conspicuously distichous phyllotaxis (Figure 4E–H). The roots were dimorphic, with short clasping roots, and elongate feeder roots that connect to the soil. Leaf development exhibited strong heteroblasty, with the smallest (earliest produced) leaves simple and entire (Figure 4A), the next set of leaves deeply pinnatifid (Figure 4B–E), and the largest leaves proximally divided and distally entire (Figure 4G), or sometimes irregularly serrate (Figure 4H). Our observations indicate that the fully entire leaves characteristic of the mature plants do not develop until after roots contact the soil (Figure 4I).

Vouchers—Northern Province, Akupe Camp, Kuriae River, Sibium Mountains, Umate Village, −9.285825 148.27266, 740 m, *James & Sundue 1503* (BISH, LAE, VT); East New Britain Province: Wild Dog Camp, former site of SINIVIT Wild Dog Mine, Collected in ravine below Wild Dog Camp, −4.62616 152.04406, 1012 m, *Sundue & Maraia 4409* (BISH, LAE, VT).

#### 2.1.5. *Crepidomanes aphlebioides* (Christ) I. M. Turner (Hymenophyllaceae)

Observations—*Crepidomanes aphlebioides* was seen growing as a short-creeping, low-trunk hemiepiphyte growing on both large (Figure 5A,B) and small (Figure 5C) trees. Plants were seen to have root dimorphy, and the feeder roots were observed to connect with the ground (Figure 5A arrows). *Crepidomanes aphlebioides* has dimorphic fronds, with short sessile (epetiolate) fronds, and larger petiolate fronds. The short fronds are highly dissected, with elongate filiform segments that spread in various directions (Figure 5C,E). In some cases, the short fronds cover the rhizome (Figure 5B). The larger fronds are planar and spread away from the rhizome (Figure 5D). Whether the development of these different frond types is related to the transition from holoepiphyte to hemiepiphyte was not determined by us.

Vouchers—East New Britain: Wild Dog Camp, former site of SINIVIT Wild Dog Mine, Omrock trail, −4.62616 152.04406, 1012 m, *Sundue & Maraia 4214* (BISH, LAE, VT).

#### 2.1.6. *Leptochilus macrophyllus* (Blume) Noot.

Observations—*Leptochilus macrophyllus* occurred as a low-trunk hemiepiphyte (Figure 6A,B) The rhizome is long creeping and dorsiventrally compressed with dorsal-lateral leaves and lateral roots (Figure 6C,D). The rhizomes also bear lateral branch buds (Figure 6C), and some older populations seemed to be spreading by branching (Figure 6A). The roots were dimorphic with short clasping roots adhering to the phorophyte, and elongate feeder roots that connected with the soil. The simple entire leaves were arching and generally spreading away from the phorophyte. Besides living trees, we also observed this species on a fallen dead branch (Figure 6A); whether the *L. macrophyllus* established on a living or dead tree was unclear to us.

Vouchers—East New Britain: Wild Dog Camp, former site of SINIVIT Wild Dog Mine, Walk to Camp III at Regess, −4.67986 152.01701, 1355 m, *Sundue & Maraia 4398* (BISH, LAE, VT).

#### 2.1.7. *Sphaerostephanos scandens* Holttum (Thelypteridaceae)

Observations—*Sphaerostephanos scandens* occurred as a low-trunk epiphyte on the root mantle of tree fern trunks (Figure 7A,B). The long-creeping rhizomes were radially symmetrical (Figure 7C), with short clasping roots and many elongate feeder roots (Figure 7D) connecting to the ground. The leaves emerged in spiral phyllotaxis and spread away from the phorophyte (Figure 7A,B,E). We found only a single population, all of which grew on tree fern trunks.

Vouchers—East New Britain: Wild Dog Camp, former site of SINIVIT Wild Dog Mine, Walk to Camp III at Regess, −4.67986 152.01701, 1355 m, *Sundue & Maraia 4420* (BISH, LAE, VT).

## 3. Discussion

Our results for the Baining Mountains indicate that at least 3% of the New Britain fern flora are hemiepiphytic. This percentage is small compared to the nearly 21% of species that were holoepiphytes, but an important finding because they represent a very different ecological niche. These new records occur in taxonomically disparate groups of eupolypod ferns—Aspleniaceae, Hymenophyllaceae, Polypodiaceae, and Thelypteridaceae; these families are not closely related to each other but show signs of morphological convergence to this habit. All species were found to have dimorphic roots (i.e., roots differentiated as either feeding or clasping roots), and all but *A. scandens* were found to have a distichous phyllotaxy (Table 2). Three of the seven species exhibited dorsiventrally flattened rhizomes, and two of them exhibited dramatic changes in leaf morphology throughout the leaf developmental series. These results reinforce the hemiepiphytic syndrome described by Testo and Sundue [5] and suggest that these traits can guide the discovery of other hemiepiphytic ferns.

Our results also reveal that both *Asplenium* and *Leptochilus* have clades comprised primarily of hemiepiphytic species. In *Asplenium*, the three species reported here were found to be closely related by Xu et al. [27]. They found a clade comprising *A. amboinsense*, *A. marattioides*, *A. scandens*, and *A. scolpendropsis*, which together were sister to other simple-leaved species, their “Neottopteris” clade. *Asplenium marattioides* has not been reported as a hemiepiphyte, but examination of *A. marattioides* herbarium specimens online (e.g., *Braithwaite 4528*, L; *Fawcett 687*, MICH; *Game 95/295*, VT) show traits characteristic of hemiepiphytes including a dorsiventral rhizome, root dimorphy, and strong morphological differences throughout the leaf developmental series. It should be targeted for future study. Species similar to *A. scolopendropsis* such as *A. schizocarpum*, and *A. translucens* should also be targeted. Xu et al. [27] also reported *A. simplicifrons* as belonging to this clade, but it is epipetric and holoepiphytic. However, the sample Xu et al. [27] used was reported to be from Papua New Guinea, where *A. simplicifrons* does not occur. Given that it was nested within two samples of *A. amboinense* in their results, it seems likely that it is a misidentified sample of that species. Ohlsen et al. [28] sampled *A. simplicifrons* from near the type locality and resolved it in an unrelated clade.

The recognition of a hemiepiphytic clade in *Asplenium* provides an opportunity to infer the evolution of habit in these clades. The hemiepiphytic clade is sister to the *Asplenium nidus* group (bird’s nest ferns) *sensu lato* in Xu et al. [27]. That clade is large and variable but is mostly epiphytic. Sister to both of these are the *A. scolopendrioides* clade and then the *A. ceterach* clade, which are terrestrial or epipetric. Although there is a great deal of disparity between these clades, the topology does indicate a transition from an ancestrally terrestrial/epipetric habit diverging into hemiepiphytic and holoepiphytic sister groups.

Our results also reveal phylogenetic patterns in *Leptochilus*. Three other species have been reported as hemiepiphytes, namely, *L. brevipes*, *L. ornithopus*, and *L. scandens*. Together, these reports suggests that hemiepiphytes might be widespread in the genus but overlooked. In addition, other evidence suggests that *Leptochilus* may often be facultatively hemiepiphytic. Yu et al. [21] reported that *L. brevipes* grows both hemiepiphytically and epipetrically, and Chen et al. [25] reported 15 species of *Leptochilus* as growing terrestrially in their phylogenetic reconstruction of habit, but several of these species are also reported as epiphytes [29]. Chen et al. [25] did not distinguish hemiepiphytic from terrestrial habits and no justification or evidence in support of the habits was provided. The discrepancy in reported habits suggest that some of those species may in fact be facultatively hemiepiphytic.

*Crepidomanes aphlebioides* was previously reported as hemiepiphytic by Hennequin et al. [8] but without documentation. Our results confirm that the species establishes on trees and then connects to the soil via elongate feeder roots. However, collection notes would indicate that at the southern edge of its range in Queensland, Australia, it grows epipetrically (*Gray 8183*, MELU; *Jones 18611*, MELU; *Ohlsen & Field s.n.* MELU). This raises the question of how and when habits shift. In their phylogenetic results, Hennequin et al. [8] resolved *C. aphlebioides* as sister to a clade of terrestrial species including *C. intermedium* (Bosch) Ebihara & K. Iwats., *C. grande* (Copel.) Ebihara & K. Iwats, and *C. thysanostomum* (Makino) Ebihara & K. Iwats. Other closely related clades comprised epiphytic species, and *Vandenboschia*, which includes several hemiepiphytes [7,8,14]. The phylogenetically clustered distribution of terrestrial and hemiephytic species in their results implies a frequent shift between these two habits evolving from an epiphytic ancestral habit.

When he described *Sphaerostephanos scandens*, Holttum [30] reported it as an epiphyte, but he may not have ever observed the species himself, and he may not have considered the distinction between holoepiphytic and hemiepiphytic habits in ferns. He did, however, describe it as climbing, which is consistent with our observations. Our interpretation that this species is a hemiepiphyte is noteworthy given that the other estimated 1200 species of Thelypteridaceae are reportedly found on soils, rocks, or in wetlands. This would suggest that *S. scandens* most likely evolved from terrestrial ancestors.

Holttum suggested that *S. scandens* is related to *S. invisus* (Forst. f.) Holttum and *S. mundas* (Rosenst.) Holttum (as *Dryopteris farinosa* Brause) and “a few other species in New Guinea which have climbing rhizomes”, but without further details. In their phylogenomic analysis, Fawcett et al. [31] supported the close relationship of *S. scandens* and a plant identified as *S*. aff. *mundas* but resolved *S. invisus* within the distantly related *Strophocaulon*. Whether a clade of hemiepiphytes remains to be discovered within *Sphaerostephanos* requires further investigation.

## 4. Materials and Methods

Field work was primarily conducted during a 2016 expedition to the Baining Mountains in East New Britain, but also draws from observations made in two other expeditions, a 2013 expedition to the Sibium Mountains in Northern (Oro) Province, and a 2014 expedition to Mt. Wilhelm in Chimbu and Madang provinces. Observations were made from one or more populations in the field and further augmented with herbarium study. When possible, all life stages were photographed in situ.

Documenting a hemiepiphytic habit is undertaken based on three distinct established criteria. The first is that gametophytes establish epiphytically on trees. Gametophyte identification can be confirmed by finding thalli persisting on juvenile sporophytes. This is conducted by searching tree trunks for successively younger sporophytes during which a developmental series of leaf morphology is discovered, allowing for the identification of the youngest sporelings. These youngest plants often retain gametophyte thali. The second criterion is that sporophytes must be observed to contact the soil via roots. This is conducted by carefully removing roots from bark of the phorophyte (host tree) and following them to the forest floor. The roots of holoepiphytes, in contrast, have roots that do not contact the soil. The third criterion required is that sporophytes never establish on the ground. This observation serves to distinguish obligate hemiepiphytes from facultative hemiepiphytes. Previous studies emphasized the need to distinguish primary hemiepiphytes from secondary hemiepiphytes; however, we agree with Zotz et al. [13] that secondary hemiepiphytes have never clearly been demonstrated for ferns or any other plant group. Therefore, “secondary hemiepiphytes” are not further considered here.

The gold standard for establishing a species as a hemiepiphyte is the positive observation of all three criteria. However, observing only the second and third criteria is sufficient in many cases. Given that sporophytes can only establish where a gametophyte had been present, the presence of a sporophyte implies where the spore germinated, and the gametophyte began growth. We believe this is a safe assumption to make for many small to medium-sized plants. However, we do not recommend making this assumption in the case of plants that have been established for long periods of time, with elongate and complicated rhizome systems such as those observed by Gay [32] for *Mickelia guianensis* (Aublet) R. C. Moran, Sundue & Labiak, or which may be climbing or scandent, as they could easily mislead observers.

Beyond these criteria, we also looked for morphological features that are expected to occur in hemiepiphytes, as described in Testo & Sundue [33]; specifically, non-cordiform (i.e., filamentous, ribbon, or strap-shaped) gametophytes, dorsiventrally flattened rhizomes, a distinctive leaf developmental series that changes with root–soil contact, and root dimorphism where the plants have distinctly short “clasping” roots that serve to attach the rhizome to the phorophyte and elongate “feeder” that descend to the ground.

Observations were made concurrently with a biological inventory of the ferns in the Sibium Mountains. Collections were opportunistic and not designed with a statistical methodology; nonetheless, we report the total number of collections and the total number of confirmed hemiepiphytes to provide a baseline quantification of the percentage of this habit in a tropical fern flora.

Specimens were identified through direct comparison of collections at BM, E, GH, K, UC, and VT, or through comparison of images available through www.pteridoportal.org. We also relied upon relevant taxonomic literature [33,34,35,36,37,38,39,40].

## 5. Conclusions

This study advances our understanding of fern ecology by expanding the taxonomic breadth and number of ferns known to grow hemiepiphytically. We found that approximately 3% of fern species in the Baining Mountains were hemiepiphytes. Our records in *Asplenium* and *Leptochilus* indicate a strong phylogenetic pattern in those genera. Additionally, our findings of hemiepiphytism in *Sphaerostephanos* (Thelypteridaceae) and *Crepidomanes* (Hymenophyllaceae) indicate that other species in those genera should be examined for further instances of hemiepiphytism.

## Figures and Tables

**Figure 1 plants-13-01104-f001:**
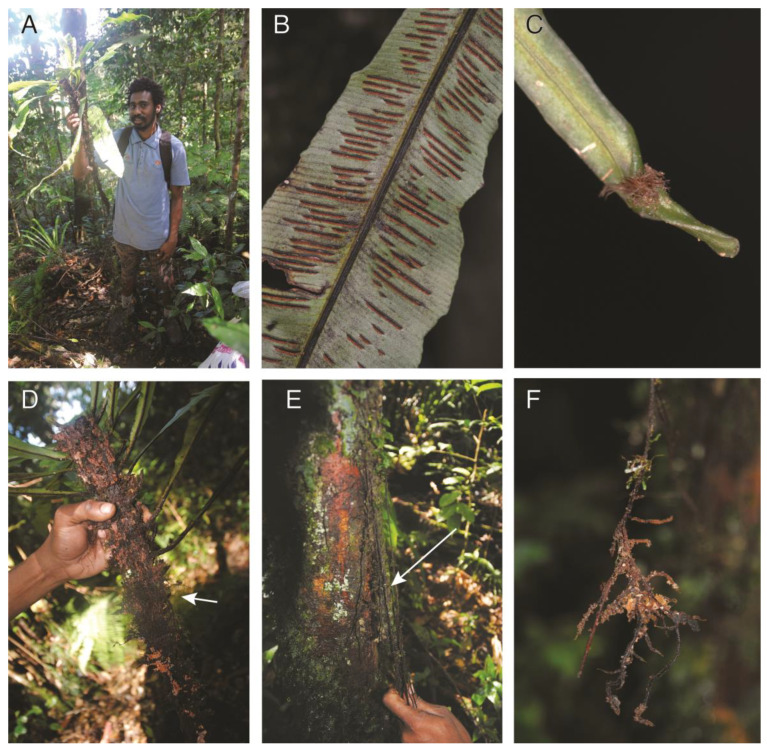
*Asplenium acrobryum*. (**A**) Sporophyte. (**B**) Sori. (**C**) Subapical proliferous bud. (**D**) Rhizome with dense clasping roots. (**E**) Descending feeder roots on the trunk base. (**F**) Feeder roots removed from soil at contact point. (**A**–**F**) *Sundue & Maraia 4250* (VT).

**Figure 2 plants-13-01104-f002:**
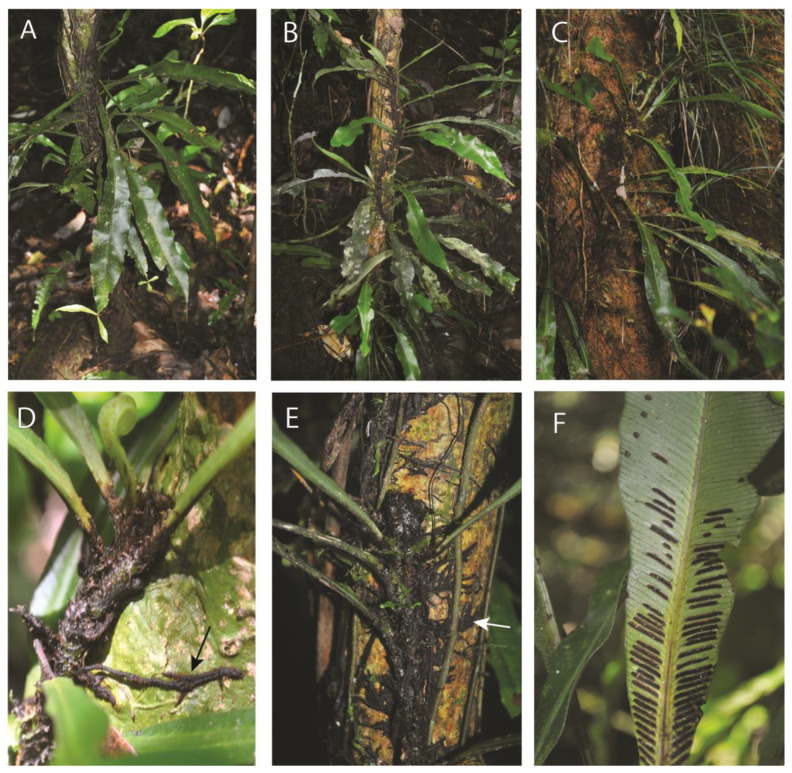
*Asplenium amboinense*. (**A**–**C**) Sporophyte growing as a low-trunk epiphyte. (**B**) Young sporophyte establishing upon the phorophyte. (**D**) Sporophyte with feeder roots (black arrow). (**E**) Dorsiventral rhizome showing leaves departing from the upper surface and clasping roots (white arrow). (**F**) Mature leaf with sori. (**A**–**F**) *James & Sundue 1519*.

**Figure 3 plants-13-01104-f003:**
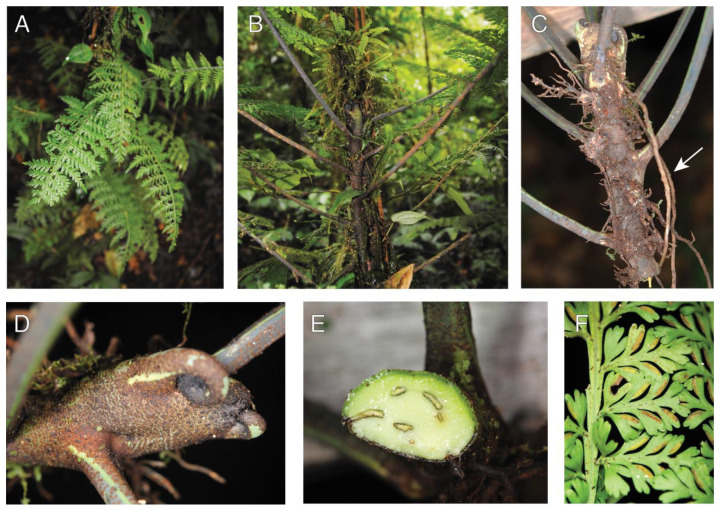
*Asplenium scandens.* (**A**) Habit of sporophyte growing as low-trunk epiphyte. (**B**) Creeping rhizome with leaves directed toward the ventral side. (**C**) Ventral surface of rhizome showing short clasping roots and elongate feeder roots (arrow). (**D**) Apex of rhizome showing phyllotaxis. The light-green longitudinal lines on the leaves are aerophores. (**E**) Rhizome cross-section showing dorsiventral compression. (**F**) Detail of sori. (**A**–**F**) *Sundue & Maraia 4333* (VT).

**Figure 4 plants-13-01104-f004:**
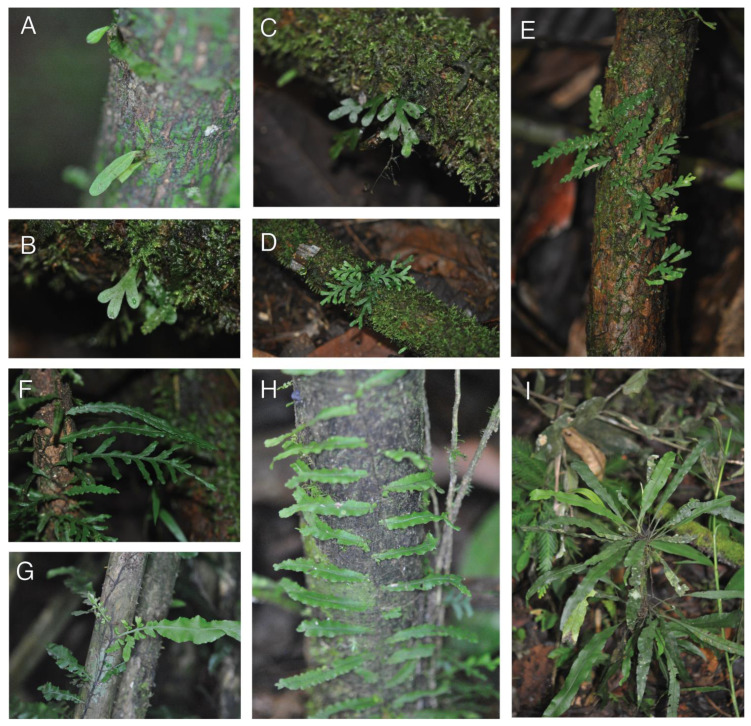
*Asplenium scolopendropsis*. (**A**) Juvenile sporophyte developing from gametophyte with first leaf. (**B**–**D**) Sporophyte showing early leaf transition series. (**E**) Small sporophyte climbing phorophyte showing distichous leaf arrangement and transition from simple to divided leaves. (**F**–**H**) Mature sporophyte showing transition from divided to simple leaves. (**I**) Mature sporophyte. (**A**–**I**) *James & Sundue 1503* (VT).

**Figure 5 plants-13-01104-f005:**
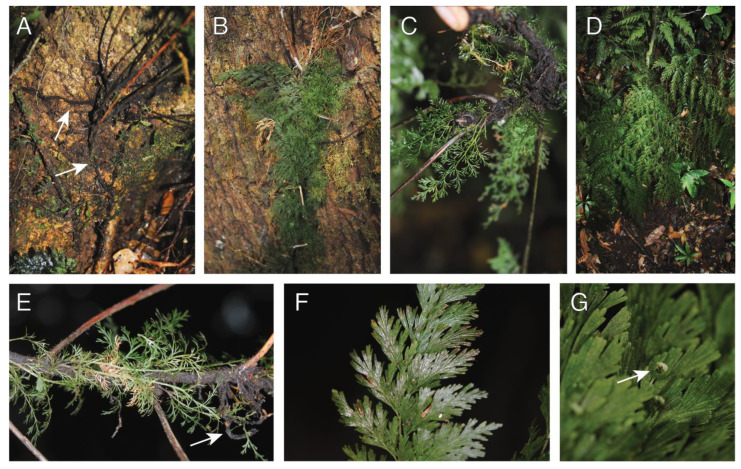
*Crepidomanes aphlebioides*. (**A**) Rhizome with elongate feeder roots growing along phorophyte (arrows). (**B**). Rhizome covered by aphlebieae (reduced leaves). (**C**) Ventral surface of rhizome with mass of roots. (**D**) Mature sporophyte growing as low-trunk epiphyte. (**E**) Rhizome with aphlebieae and clasping roots (arrow); apex (not seen) toward the right. (**F**) Detail of frond. (**G**) Detail of frond with receptacle. (**A**–**G**) *Sundue & Maraia 4214* (VT).

**Figure 6 plants-13-01104-f006:**
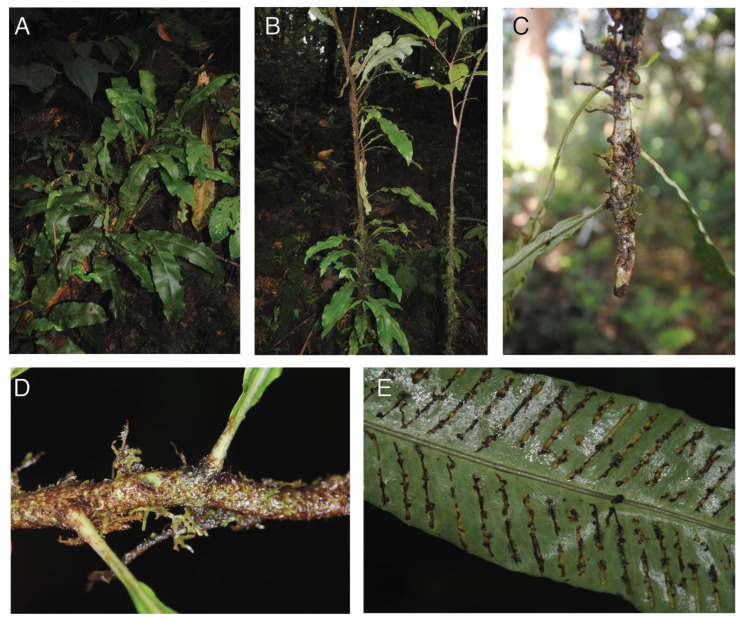
*Leptochilus macrophyllus*. (**A**,**B**) mature sporophyte growing as low-trunk epiphyte. (**C**) Ventral surface of dorsiventrally compressed rhizome with lateral clasping roots. (**D**) Dorsal surface of rhizome with distichous phyllotaxis. (**E**) Mature sori. (**A**–**E**) *Sundue & Maraia* 4398 (VT).

**Figure 7 plants-13-01104-f007:**
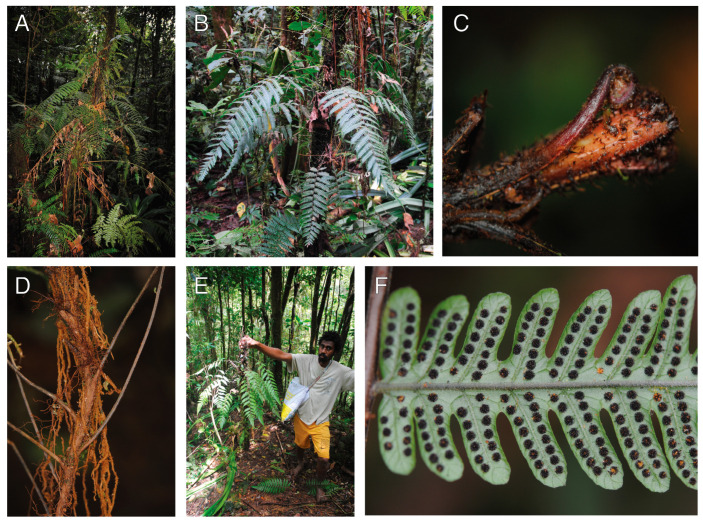
*Sphaerostephanos scandens*. (**A**,**B**) Sporophyte growing as a low-canopy epiphyte upon a tree fern trunk. (**C**) Rhizome apex showing radially symmetrical growth. The light-colored longitudinal lines are aerophores. (**D**) Rhizome with petioles departing petioles and elongate feeder roots. (**E**) Sporophyte removed from the phorophyte. (**F**) Abaxial lamina surface with mature sori (arrow). (**A**–**F**) *Sundue & Maraia 4420* (VT).

**Table 1 plants-13-01104-t001:** Habit criteria scored for the seven species of hemiepiphytes found in this study. Check marks indicate that the observation was confirmed.

Species	Plants Are Low-Trunk Epiphytes	Plants Never Seen Growing Terrestrially	Gametophytes or Young Plants Observed Upon Phorophyte	Roots Connected to Ground
*Asplenium acrobryum*	✓	✓	✓	✓
*Asplenium amboinense*	✓	✓		✓
*Asplenium scandens*	✓	✓		✓
*Asplenium scolopendropsis*	✓	✓	✓	✓
*Crepidomanes aphlebioides*	✓	✓		✓
*Leptochilus macrophyllus*	✓	✓		✓
*Sphaerostephanos scandens*	✓	✓		✓

**Table 2 plants-13-01104-t002:** Observed morphological characters of the hemiepiphytic habit. Check marks indicate that the observation was confirmed, and “no” indicates it was confirmed to not be the case.

Species	Gametophytes Non-Cordiform	Rhizomes Dorsiventrally Flattened	Roots Dimorphic with “Clasping” and “Feeder” Roots	Leaves with a Developmental Series That Changes upon Contact with the Ground	Phyllotaxis Distichous or Leaves All Emerging from One Side
*Asplenium acrobryum*		✓	✓		✓
*Asplenium amboinense*		✓	✓		✓
*Asplenium scandens*		✓	✓		no
*Asplenium scolopendropsis*	✓	no	✓	✓	✓
*Crepidomanes aphlebioides*		no	✓		✓
*Leptochilus macrophyllus*		✓	✓		✓
*Sphaerostephanos scandens*		no	✓		✓

## Data Availability

Original photographs for all collections are freely available via www.fernsoftheworld.com.

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
