# Peer review of "Field Work in Papua New Guinea Documents Seven New Records of a Hemiepiphytic Habit in Ferns"

_plants, 2024, doi:10.3390/plants13081104_

Round 1

Reviewer 1 Report

Comments and Suggestions for Authors

A great study that increases our knowledge of the diverse but still poorly known and documented New Guinean fern flora. I have included below some notes on how I think the manuscript can be improved. The most important thing that needs attention is the paragraph in the discussion regarding A. simplicifrons. I think this correction also makes the findings of the study more significant because I think that this Asplenium clade will be entirely hemiepiphytic and possibly the only clearly observable synapomorphy in an otherwise quite morphologically variable clade.

Line 17: macrophyllous should be macrophyllus throughout manuscript i.e. line 53, 64, 192, 200, and 244.

Line 39: new 10 should be ten new

Line 40: Perhaps define in parentheses what is meant by holoepiphyte given that hemiepiphytic is defined above.

Line 62: A. scandens should be italicised.

Line 63: Dorsiventrally should not be capitalised.

Figure 2. There doesn’t seem to be arrows given in part E or F.

Line 192: L. macrophyllous should be L. macrophyllus and italicised.

Line 233: I think there is a good possibility this Asplenium clade is entirely hemiepiphytic. The A. simplicifrons sample included in Xu et al. (2020) is misidentified. It is most likely either A. acrobryum or A. amboinense. The true position of A. simplicifrons is given in Ohlsen et al. (2014) Australian Systematic Botany 27: 355-371, as closely related to A. subemarginatum (a different clade in Xu et al. 2020). The A. simplicifrons sample used in Ohlsen et al. was collected from wild material on the Atherton Tableland where this species occurs. The sample from Xu et al. comes from cultivated material in PNG where A. simplicifrons does not occur and probably isn’t grown. This paragraph needs to be changed to reflect this information. The species other than A. scolopendrioides of the former Diplora segregate i.e. Asplenium schizocarpum and A. translucens, while not sampled yet in molecular studies will almost certainly belong to this clade because of their paired and facing sori unique among Asplenium. Perhaps these along with A. marattioides should also be targeted for future study.

Line 242: Perhaps it is worth discussing that it is unclear what habit hemiepithytism is derived from. In the case of Asplenium, the hemiepiphytic clade was not nested in a clearly terrestrial or epiphytic clade so it is not possible to determine likely evolutionary transition based on phylogeny of extant species.

Line 257: I think you have demonstrated that it can establish on trees and connect to soil. In Australia Crepidomanes aphlebioides occurs on rock in steep creek embankments with seemingly no interaction with trees (details in some of these records: https://avh.ala.org.au/occurrences/search?q=taxa%3A%22Crepidomanes%20aphlebioides%22&qualityProfile=ALA&qc=data_hub_uid%3Adh9&fq=country%3A%22Australia%22). I think this is a reasonably convincing example of a facultative hemiepiphyte.

Line 273: H. scandens should be S. scandens.

Methods: How were the species identified? I think sources used for identification if used should be stated or if identification was done based on comparison to types/other herbarium collections this should be mentioned. I have difficulties separating Asplenium acrobryum and A. amboinense from each other based on herbarium material. I have not seen them in the wild so perhaps it is more obvious in the wild. I do think it would be a worthwhile addition to mention how the authors distinguish A. acrobryum from A. amboinense.

Comments on the Quality of English Language

Only a few minor typos detected that I have outlined in the notes above.

Author Response

Thanks for your comments. These were some of the most helpful comments I can recall receiving. They have greatly added to the paper and we really appreciate that. 

We made all the the corrections suggested. Thanks for pointing those out. 

Knowing that the story behind the Asplenium simiplicifrons sample in Xu was a great help. We also gained a lot by knowing that Crepidomanes aphlebioides is epipetric in Queensland. We added to the discussion on these topics. 

Regarding the distinction between A. amboinense and A. acrobryum. We think they best characters are sorus width and length. We agree the two species are very similar. At the time when we identified them , we had no problem distinguishing them, but that was several years ago and we do not have the material in front of us at the moment. Further study may demonstrate that they are conspecific, but that would not invalidate the results here. Since there are no recent (or any?) publications that challenge their distinction, and since many taxonomists have maintained them as distinct, we chose to do so for now. Also, because this is not a taxonomic paper, it would seem like an odd place to tackle this question. However, this question will stick with us, and we will take a closer look when we have a chance. One of us (MS) will be at the VT herbarium in a few months and can look into this more then. 

Reviewer 2 Report

Comments and Suggestions for Authors

This is a straightforward paper documenting the hemiepiphytic habit in 7 fern species. It is well researched, the species are well documented, and the conclusions are well supported by the data. Overall, the topic of hemiephytism in ferns in highly interesting and poorly understood, so this study adds important information. I only have a few minor comments:

38: should it be gametophytes?

63: Delete Careful examination of the?

72: place statement in a spatial context, e.g., of fern and lycophyte species diversity in the study regions. (many other options are feasible)

192: species name in italics

224: do species represent an ecological niche? I would say the occupy it …

380: Botany

Author Response

Thanks for your careful review. We made all of the changes. These edits were very helpful.